# Ouabain and Digoxin Activate the Proteasome and the Degradation of the ERα in Cells Modeling Primary and Metastatic Breast Cancer

**DOI:** 10.3390/cancers12123840

**Published:** 2020-12-19

**Authors:** Claudia Busonero, Stefano Leone, Fabrizio Bianchi, Elena Maspero, Marco Fiocchetti, Orazio Palumbo, Manuela Cipolletti, Stefania Bartoloni, Filippo Acconcia

**Affiliations:** 1Department of Sciences, Section Biomedical Sciences and Technology, University Roma Tre, Viale Guglielmo Marconi, 446, I-00146 Rome, Italy; claudia.busonero@uniroma3.it (C.B.); stefano.leone@uniroma3.it (S.L.); marco.fiocchetti@uniroma3.it (M.F.); manuela.cipolletti@uniroma3.it (M.C.); stefania.bartoloni2@uniroma3.it (S.B.); 2Cancer Biomarkers Unit, Fondazione IRCCS Casa Sollievo della Sofferenza, 71013 San Giovanni Rotondo (FG), Italy; f.bianchi@operapadrepio.it; 3Fondazione Istituto FIRC di Oncologia Molecolare (IFOM), 20139 Milan, Italy; elena.maspero@ifom.eu; 4Division of Medical Genetics, Fondazione IRCCS Casa Sollievo della Sofferenza, 71013 San Giovanni Rotondo (FG), Italy; o.palumbo@operapadrepio.it

**Keywords:** 17β-estradiol, estrogen receptor, breast cancer, proteasome, ouabain, digoxin

## Abstract

**Simple Summary:**

Breast cancer (BC) treatment relies on the detection of the estrogen receptor α (ERα). ERα-expressing BC patients are treated with anti-estrogen drugs (i.e., tamoxifen and fulvestrant). Despite their proven efficacy, these drugs cause serious side effects in a significant fraction of the patients, including both tumor insurgence in secondary organs, and resistant phenotypes, which result in a relapsing disease with scarce treatment options. Thus, new drugs for treatment of primary and metastatic BC (MBC) are needed. Here, we report the characterization of two cardiac glycosides (CGs) (i.e., ouabain and digoxin), approved by the FDA for treatment of heart disease, as novel ‘anti-estrogen’-like drugs. We found that these drugs induce ERα degradation, and prevent the proliferation of cellular models of primary and metastatic BC cells. Remarkably, we discovered that these CGs are activators of the proteasome, and therefore may be repurposed for treatment not only of BC, but also for other proteasome-based diseases.

**Abstract:**

Estrogen receptor α expressing breast cancers (BC) are classically treated with endocrine therapy. Prolonged endocrine therapy often results in a metastatic disease (MBC), for which a standardized effective therapy is still lacking. Thus, new drugs are required for primary and metastatic BC treatment. Here, we report that the Food and Drug Administration (FDA)-approved drugs, ouabain and digoxin, induce ERα degradation and prevent proliferation in cells modeling primary and metastatic BC. Ouabain and digoxin activate the cellular proteasome, instigating ERα degradation, which causes the inhibition of 17β-estradiol signaling, induces the cell cycle blockade in the G2 phase, and triggers apoptosis. Remarkably, these effects are independent of the inhibition of the Na/K pump. The antiproliferative effects of ouabain and digoxin occur also in diverse cancer models (i.e., tumor spheroids and xenografts). Additionally, gene profiling analysis reveals that these drugs downregulate the expression of genes related to endocrine therapy resistance. Therefore, ouabain and digoxin behave as ‘anti-estrogen’-like drugs, and are appealing candidates for the treatment of primary and metastatic BCs.

## 1. Introduction

Breast cancer (BC) is the most common cause of cancer-related death among women of all ages worldwide. The most frequent type of BC is invasive ductal carcinoma, which is estrogen receptor α-positive (ERα+) in most cases (70%) [1]. 17β-Estradiol (E2) signals through the ERα and promotes BC progression via integrated stimulation of transcriptional and non-transcriptional ERα activity [2]. Thus, the clinical approach for ERα+ tumors consists of endocrine therapy [1], aiming to inhibit proliferative E2:ERα signaling. Endocrine therapy drugs either reduce E2 availability (e.g., aromatase inhibitors (AIs)) or block ERα activity by directly binding to the receptor; 4OH-tamoxifen (4OH-Tam), a selective ER modulator (SERM), inactivates ERα transcriptional activity, and fulvestrant (ICI182,780-ICI), a selective ER downregulator (SERD), eliminates ERα from BC cells and prevents receptor transcriptional functions [1]. Thus, endocrine therapy restrains BC progression.

Nonetheless, endocrine therapy has substantial limitations: AIs are typically reserved for postmenopausal women with BC, and produce musculoskeletal failure, while 4OH-Tam and ICI have severe side effects (e.g., endometrial cancer for 4OH-Tam), and often cause cell resistance [1]. 4OH-Tam insensitive tumors often still express the ERα, with some of them being resistant at the time of the diagnosis (i.e., de novo resistant), while others acquire 4OH-Tam insensitivity during prolonged 4OH-Tam treatment through various mechanisms, including the selection of tumor cells that express a mutated constitutively active ERα variant (e.g., mutation of the receptor tyrosine (Y) 537 into serine (S)) [2,3,4,5,6]. These tumors have proliferative advantages that result in a relapse of the disease and in the recurrence of a metastatic BC (MBC), for which only limited pharmacological options exist [1].

Consequently, the current challenge is to complement and/or modify the BC drugs available to fight ERα+ primary and MBC. Different lines of research are taking place to identify novel SERMs (e.g., basedoxifene) [7] and/or SERDs (e.g., AZD9496 and GDC-9545) [2,8,9,10], or to identify new pathways that could be targeted alone or in combination with endocrine therapy drugs (e.g., CDK4/CDK6 or CDK7 inhibitors) [2,11,12,13,14,15,16], especially for MBCs.

Recently, we proposed a strategy to expand the repertoire of effective compounds against ERα+ BC by unveiling a shortcut to rapidly identify new anti-ERα+ BC drugs. In this respect, we introduced the notion that drugs, which do not necessarily bind to ERα but rather change the receptor protein amount in BC cells, can prevent BC cell proliferation and subsequently tumor growth [2]. In turn, we used the ERα levels in BC cells as a novel pharmacological target to screen a library of 1018 Food and Drug Administration (FDA)-approved drugs with the aim of identifying the compounds that can modify ERα protein levels and inhibit BC cell proliferation [2,17,18,19]. This approach identified medications with potential ‘anti-estrogen’-like effects [17,18,19].

Among those drugs we also identified two cardiac glycosides (CGs), i.e., ouabain-OU and digoxin-Digo [20], as compounds that potentially reduce ERα levels, and prevent cell proliferation in ductal carcinoma cells (MCF-7 cells) [18]. OU and Digo are established plant-derived drugs that inhibit the activity of the Na/K ATPase, and are used in the treatment of congestive heart failure (i.e., digitalis treatment) [21].

Here, we evaluated and characterized the effects of these CGs in different models of primary and MBC, and found that OU and Digo are antiproliferative agents, functioning as small molecule activators of the cellular proteasome.

## 2. Materials and Methods

### 2.1. Cell Culture and Reagents

17β-estradiol (E2), DMEM (Dulbecco’s Modified Eagle Medium, with and without phenol red), and fetal calf serum were purchased from Sigma-Aldrich (St. Louis, MO, USA). Bradford protein assay kit as well as anti-mouse and anti-rabbit secondary antibodies were obtained from Bio-Rad (Hercules, CA, USA). Antibodies against ERα (F-10 mouse), ubiquitin (Ub) (P4D1 mouse), p53 (DO-1 mouse), Bcl-2 (C2 mouse), cyclin D1 (H-295 rabbit), UBE1 (2G2 mouse), cathepsin D (H75 rabbit), pS2 (FL-84 rabbit), SREBP1 (H160 rabbit) were obtained from Santa Cruz Biotechnology (Santa Cruz, CA, USA); anti-vinculin and anti-tubulin antibodies were purchased from Sigma-Aldrich (St. Louis, MO, USA). Anti-SREBP2 (ab28482 rabbit) was purchased from Abcam (Cambridge, MA, USA). Anti-PARP antibody (95425 rabbit) was purchased by Cell Signaling Technology (Danvers, MA, USA). Chemiluminescence reagent for Western blotting was obtained from BioRad Laboratories (Hercules, CA, USA). Faslodex (i.e., fulvestrant or ICI182,780), AZD9496, GDC-9545, Mg-132 (Mg), and 4OH-tamoxifen (4OH-Tam) as well as the 20S recombinant proteasome were purchased from Tocris (USA). All the other products were obtained from Sigma-Aldrich. Analytical- or reagent-grade products were used without further purification. The identities of all the used cell lines (i.e., MCF-7, ZR-75-1, MCF10a-ATTC, LGC Standards S.r.l., Milano, Italy as well as Y537S and Tam Res cells [2,15,22]) were verified by STR analysis (BMR Genomics, Padua, Italy).

### 2.2. Western Blotting Analyses

Cells were grown in 10% serum and DMEM with phenol red, except for the experiments in which E2 was administered. For the E2 experiments, 24 h before E2 administration, the growth medium was replaced with 1% charcoal-stripped fetal calf serum medium in DMEM without phenol-red, and cells were left untreated for 24 h. Subsequently, cells were stimulated with E2 at the indicated time points. Cells were lysed in Yoss Yarden (YY) buffer (50 mM HEPES at pH 7.5, 10% glycerol, 150 mM NaCl, 1% Triton X-100, 1 mM EDTA, 1 mM EGTA) plus protease and phosphatase inhibitors, except for PARP detection for which cells were lysed in Tris HCl 0.125 M pH 6.8. Western blotting analyses were performed by loading 20–30 µg of protein on SDS-gels. Gels were run and transferred into nitrocellulose membranes with Biorad Turbo-Blot semidry transfer apparatus. Immunoblotting was carried out by incubating membranes with 5% milk (60 min), followed by overnight incubation with the indicated antibodies. Secondary antibody incubation was performed for an additional 60 min. Bands were detected using a Biorad ChemiDoc^TM^ apparatus. The original whole blot data is shown in Source data file of Appendix A.

### 2.3. Cell Proliferation and Cell Cycle Assays

For growth curves (Figure 1A,B and Appendix A) and drug synergy studies, the xCELLigence DP system (ACEA Biosciences, Inc., San Diego, CA) Multi-E-Plate station was used to measure the time-dependent response to the indicated drugs, using real-time cell analysis (RTCA), as previously reported [23,24]. Briefly, the number of cells (i.e., normalized cell index) is directly proportional to the measured electric impedance of the cells on the well surface. MCF-7, ZR-75-1, Tam Res, and Y537S ERα-expressing MCF-7 (Y537S) cells were seeded in E-Plates 96 in growth media. After overnight monitoring of growth at 15 min intervals, drugs were added according to the following protocol: 4OH-tamoxifen, fulvestrant (ICI182,780), AZD9496 and GDC-9545 at 10, 100, 1000, 100,000 nM separately, or in combination with ouabain or digoxin at concentrations of 30, 60, 100, 250 nM. Cells remained in the medium until the end of the experiment. Cellular responses were then recorded every 15 min, for a total period of 5 days. Next, synergy index was calculated with Combenefit freeware software [25]. The normalized cell index at 5 days for the combination of drugs at the lowest dose that exhibited a synergistic effect on cell proliferation was obtained, and is presented in Figure 9A,B.

For cell cycle analysis, after each treatment, 1 × 10^6^ cells were washed twice with PBS, fixed dropwise with ice cold ethanol (70%), and rehydrated with PBS. DNA staining was performed by incubating cells for 30 min at 37 °C in PBS, containing 0.18 mg/mL propidium iodide (PI) and 0.4 mg/mL DNase-free RNase (type 1-A). Samples were measured with a CytoFlex Flow Cytometer (Beckman Coulter). Cell cycle analysis was performed using a CytExpert v.2.4 software (Beckman Coulter). Doublet discrimination was performed by an electronic gate on FL3-Area vs. FL3-Height parameters. For hypodiploid peak analysis, Nicoletti’s protocol was followed [26]. Briefly, cell pellet was resuspended in 500 µL of PBS + 500 µL of hypotonic staining solution (0.1% sodium citrate (*w*/*v*), 0.1% Triton X-100 (*v*/*v*), 50 µg/mL propidium iodide, pH 7.8). Cells were incubated for 30 min at room temperature. Finally, 20,000 total events, presented in a logarithmic scale were acquired directly in the staining buffer, and percentage of the hypodiploid peak was calculated by a proper electronic marker.

### 2.4. Measurement of Proteasome Activity

Proteasome activity was measured by employing the Proteasome-Glo™ Assay Kit for chymotrypsin-, trypsin-, and caspase-like activities, purchased from Promega (Madison, MA, USA). Each one of the following experiments was performed at least in triplicates. Briefly, for measurements of proteasome activity in cells, the indicated cell lines were seeded in 96-well plates (10,000 cells/well; each condition in triplicates) in growth media, for 24 h. Next, cells were treated with different doses of ouabain, digoxin, or equal quantities of vehicle (DMSO) only for digoxin, for an additional 8 or 24 h. Notably, the proteasome inhibitor Mg-132 (1 µM) was used as internal control for all assays. After treatment, the three activities of the proteasome were measured according to manufacturer’s instructions, in a Tecan-Spark Elisa reader every other 30 s, for a total period of 30 min. For in vitro measurements of proteasome activity, 30 µg of recombinant 20S proteasome (Boston Biochem, Boston, MA, USA) was placed in contact with ouabain alone, or in the presence of different doses of the proteasome inhibitor Mg-132 (the doses for calculating the inhibitor concentration 50 (IC_50_) were adjusted for each activity, according to previously reported data [27]), immediately before measuring the three activities of the proteasome. This was done according to the kit manufacturer’s instructions, in a Tecan-Spark Elisa reader set at 37 °C, every other 30 s for a total period of 30 min. The data obtained within the first 10 min of the measurements were used for this study.

### 2.5. Measurement of Na/K ATPase Activity

Cells were seeded in 96-well plates (10,000 cells/well; each condition in a quadruplicate) in growth media for 24 h. After 24 h, the Na/K ATPase assay was performed as previously reported [28]. Briefly, cells were washed with NaCl (0.9%) and lysed in ddH_2_O. Then, cells were incubated for 10 min at 37 °C in Assay Buffer 2× (36 mM histidine, 36 mM imidazole, 160 mM NaCl, 30 mM KCl, 6mM MgCl_2_, 0.2 mM EGTA, pH 7.1) in the presence or absence of different doses of ouabain (10^−9^ to 10^−2^ M), digoxin (10^−9^ to 10^−4^ M), or equal quantities of vehicle (DMSO) only for digoxin, for an additional 24 h. Next, 1 mM ATP was added to the reaction mixture, and the plates were incubated for 24 h at 37 °C in 5% CO_2_. The following day, the reaction was blocked using 25 µL/well of SDS 5%, and 125 µL/well of colorimetric solution (ammonium molybdate/H_2_SO_4_ followed by ascorbic acid) was added, in order to measure the concentration of free P_i_ in each sample. The Na/K ATPase activity was derived by subtracting the activity measured in the presence of ouabain from the total activity. Numeric values of absorbance were obtained with a Tecan-Spark Elisa reader, with the wavelength set at 630 nm. All experiments were performed at least in triplicates.

### 2.6. ERα Binding Assay

A fluorescence polarization (FP) assay was used to measure ouabain, digoxin, and 17β-estradiol binding affinity to recombinant ERα in vitro as previously described [29].

### 2.7. RNA Isolation and qPCR Analysis

The sequences for gene-specific forward and reverse primers were designed using the OligoPerfect Designer software program (Invitrogen, Carlsbad, CA, USA). The primers used for human ERα were 5′-GTGCCTGGCTAGAGATCCTG-3′ (forward) and 5′-AGAGACTTCAGGGTGCTGGA-3′ (reverse), and for human GAPDH were 5′-CGAGATCCCTCCAAAATCAA-3′ (forward) and 5′-TGTGGTCATGAGTCCTTCCA-3′ (reverse). Total RNA was extracted from the cells using TRIzol Reagent (Invitrogen, Carlsbad, CA, USA), according to the manufacturer’s instructions. To determine gene expression levels, cDNA synthesis and qPCR were performed using the GoTaq 2-step RT-qPCR system (Promega, Madison, MA, USA), with an ABI Prism 7900HT Sequence Detection System (Applied Biosystems, Foster City, CA, USA), according to the manufacturer’s instructions. Each sample was tested in triplicates, the experiment was repeated twice, and gene expression was normalized to GAPDH mRNA levels.

### 2.8. Affimetrix Analysis

Total RNA was extracted using RNeasy kit (Qiagen, Hilden, Germany), according to manufacturer’s protocol, and was quantified using a NanoDrop 2000 system (Thermo Scientific, Waltham, MA, USA). A GeneChip Pico Reagent Kit (Affymetrix, Santa Clara, CA, USA) was used to amplify 5 ng of total RNA, according to the manufacturer’s protocol. Quality control of the RNA samples was performed using an Agilent Bioanalyzer 2100 system (Agilent Technologies, Santa Clara, CA, USA). Gene expression profiling was performed using the Affymetrix GeneChip^®^ Human Clariom S Array (Thermo Fisher Scientific), including more than 210,000 distinct probes representative of 21,448 annotated genes (hg19; Genome Reference Consortium Human Build 37 (GRCh37); https://www.ncbi.nlm.nih.gov/assembly/GCF_000001405.13/). RNA samples were amplified, fragmented, and labeled for array hybridization according to manufacturer’s instruction. Samples were then hybridized overnight, washed, stained, and scanned using the Affymetrix GeneChip Hybridization Oven 640, Fluidic Station 450 and Scanner 3000 7G (Thermo Fisher Scientific), to generate raw data files (.CEL files). Quality control and normalization of Affymetrix. CEL files were performed using the TAC software (v4.0; Thermo Fisher Scientific), by performing the “Gene level SST-RMA” summarization method with human genome version hg38 (https://www.ncbi.nlm.nih.gov/assembly/GCF_000001405.26/). Gene expression data were log2 transformed before analyses. Class comparison analysis for identifying differentially regulated genes was performed using a t-test with a random variance model and 1000 random data permutations to compute the local false discovery rate (FDR or *q-*value). An FDR <0.05 was used as cutoff for significantly regulated gene selection. Hierarchical clustering and heatmaps analyses were performed using Cluster 3.0 (http://bonsai.hgc.jp/~mdehoon/software/cluster/software.htm) and Java Tree View (http://jtreeview.sourceforge.net). The uncentered correlation and centroid linkage clustering method was adopted. Ingenuity pathway analysis (IPA; QIAGEN, Hilden, Germany) was performed to identify canonical pathways enriched in ouabain/digoxin regulated genes. Significantly enriched pathways were defined as those with a *q*-value (Benjiamini and Hochberg correction) of less than 0.05. IPA was also used to perform Upstream Regulator analysis for identification of upstream transcriptional regulators (TR) that can explain the observed changes in gene expression. Briefly, for each potential TR, two statistical measures, an overlap *p*-value and an activation z-score are computed. The overlap *p*-value calls potential TR, based on significant overlap between the set of genes (i.e., ouabain/digoxin regulated) and known targets regulated by a TR. The activation z-score is used to infer likely activation states of TR, based on comparison with a model that assigns random regulation directions.

### 2.9. Dot Blot Analysis

For dot blot analyses, cells were treated as described above, and subsequently harvested and lysed in YY buffer. Lysates (10–15 µg) were diluted in a final volume of 100 µL with YY buffer, and mechanically transferred onto a nitrocellulose membrane with a Bio-Dot^®^ Microfiltration apparatus (Biorad). The filter was then stained with Ponceau Red staining (Sigma-Aldrich), de-stained with ddH_2_O, and the resulting image was acquired with a digital scanner. Next, the nitrocellulose membrane was treated as a regular Western blotting (see above).

### 2.10. Tumor Spheroid Formation

Tumor spheroid formation was performed as previously reported [30]. Briefly, MCF-7 and Y537S ERα-expressing MCF-7 (Y537S) cells were seeded (10,000 cells/well) in ultra-low attachment surface 24-well-plates (Sigma-Aldrich), with 1 mL/well in growing condition, for 48 h. Next, using an optical microscope, images were obtained for each untreated well (i.e., prior to treatment, time 0). Next, cells were treated in quadruplicate with ICI182,780 (100 nM), ouabain (100 nM), digoxin (100 nM), and with vehicle (DMSO). After 48 h, the cell culture medium was replaced, using a 70 µm nylon sterile cell strainer for each condition, in order to maintain spheroids with a diameter greater than 70 µm, and to remove dead cells and spheroids with diameters smaller than 70 µm. Subsequently, the treatment was repeated. Seven days post initial drug administration, images were obtained for each well. Number of spheroids was quantitated using the freeware software Image J.

### 2.11. Transthiolation Assay

Thioester formation was performed in two steps. First, an incubation reaction was performed using Ube2D3 (4 μM), biotinylated Ub (8 μM BML-UW8705, Enzo Lifescience, Ann Arbor, MI, USA), and E_1_ enzyme (150 nM), for 2 h at room temperature in ubiquitination buffer, in the presence and in the absence of OU at the indicated concentrations. In the second step, ATP (4 mM) was added to the reaction, and the mix was moved to 37° C. Thioester formation on the E_1_ and E_2_ was monitored by quenching the reaction after 15 min with Laemmli buffer, with and without a reducing agent (DTT, 100 mM). Samples were loaded on gradient gel (4–20% Biorad), transferred to a nitrocellulose membrane, and detection was obtained by immunoblot, using HRP conjugated streptavidin (Thermo Scientific).

### 2.12. Zebrafish Xenografts

Zebrafish xenograft experiments were performed by ZeClinics (Barcelona, Spain) according to the procedure published in [31], with modifications in image analysis to allow measurement of the tumor area, rather than tumor volume. Briefly, Casper adult Zebrafish (*Danio Rerio*) were mated to obtain embryos. The study was performed under the ethical approval code 10567, provided by the Generalitat of Catalunya. Forty-eight hours post fertilization, larvae were injected with 200–400 Y573S cells in the perivitelline space. An hour later, noninjected and nonspecifically injected larvae were discarded. For the experiment in Figure 9D, xenografts were imaged at timepoint 0 (i.e., 24 h post injection) and at time point 2 (i.e., 96 h post injection and 3 days drug treatment). Tumor development was evaluated by comparing the cell masses and dissemination between time point 0 and time point 4 for each larva. Ratio of areas between timepoints 0 and 2 were calculated for each larva and mean primary tumor areas average fold change between the two time points was calculated both for control- and drug-treated groups. The relative final report containing the details of the procedure used and the relative analysis is available to share upon reasonable request.

### 2.13. Statistical Analysis

A statistical analysis was performed using the InStat version 8 software system (GraphPad Software Inc., San Diego, CA). Densitometric analyses were performed using the freeware software Image J, by quantifying the band intensity of the protein of interest with respect to the relative loading control band (i.e., vinculin or tubulin) intensity. The *p* values and the applied statistical test are provided in the figure captions.

## 3. Results

### 3.1. Evidence of Na/K ATPase-Independent Effects of Ouabain and Digoxin in Breast Cancer Cells

OU and Digo dose response curves were obtained to find both the effective dose 50 (ED_50_) required to reduce ERα content, and the inhibitor concentration 50 (IC_50_) required to prevent cell proliferation. The experiments were performed in cell lines modeling either primary ERα+ breast tumors sensitive to 4OH-Tam (i.e., MCF-7 and ZR-75-1 cells) or modeling metastatic tumors with acquired resistance to 4OH-Tam (i.e., 4OH-Tam resistant MCF-7 cells–Tam Res) [22] or with genetic resistance to endocrine therapy drugs (i.e., MCF-7 cells CRISPR/CAS9 engineered to express the Y537S ERα point mutant-Y537S) [2,15]. In all tested cell lines, both OU and Digo induced ERα degradation within 24 h, in a dose-dependent manner with a calculated ED_50_ in the range of 50–100 nM for OU, and 50–400 nM for Digo (Figure 1A,B and Appendix A). In addition, OU and Digo reduced the proliferation of BC cells in a cell type specific manner, with a calculated IC_50_ at 5 days for OU of ≈50 nM, and 60–200 nM for Digo (Figure 1A,B and Appendix A). These data confirm that OU and Digo induce ERα degradation, and prevent proliferation in cells modeling primary and 4OH-Tam resistant BC cells.

As OU and Digo inhibit the Na/K ATPase enzyme [21], we next examined whether the reduction in ERα intracellular levels could be attributed to a blockade of the Na/K pump activity in BC cells. Thus, we measured the Na/K ATPase activity in all the above-mentioned BC cells in the presence of various doses of OU and Digo, for 24 h. Surprisingly, while, as expected, the CGs inhibited the Na/K ATPase activity in all tested cell lines [21], they did not inhibit the activity of the Na/K pump in the ZR-75-1 (Figure 1A,B). Moreover, we noticed that the IC_50_ for the inhibition of the Na/K ATPase enzyme for OU was higher in all BC cells than both the ED_50_ required to reduce ERα content, and the IC_50_ required to prevent cell proliferation (Figure 1A,B). The same phenomenon was observed for Digo in Y537S, but not in MCF-7 and Tam Res cells (Figure 1A,B). Therefore, these data, together with the insensitivity of the Na/K pump to both OU and Digo, expressed in ZR-75-1 cells, suggest that the observed OU and Digo effects on ERα levels and cell proliferation in BC cells do not occur entirely in parallel with the inhibition of Na/K ATPase enzyme.

To further investigate this issue, we used the ZR-75-1 cells in which the Na/K ATPase enzyme was not inhibited by OU and Digo. In this cell line, we performed cell cycle analysis, and measured the apoptotic induction after treatment with different doses of OU and Digo. As shown in Figure 1C,C’, the CGs increased the percentage of the cells in the G2 phase of the cell cycle in ZR-75-1 cells, as well as in other BC cells, within 24 h (Appendix A). In line with the observed CGs-induced blockade in the G2 phase of the cell cycle, treatment of ZR-75-1 cells with OU and Digo for 72 h induced a significant increase in the abundance of the cleaved PARP (Figure 1D,E), and in the sub-G1 phase of the cell cycle (Figure 1F,F’).

Therefore, OU and Digo reduce ERα levels, and establish a cell cycle block in the G2 phase, accompanied by an induction of apoptotic cell death. Remarkably, these effects occur independently of the inhibition of the Na/K ATPase enzyme in ZR-75-1 cells.

### 3.2. The Ouabain and Digoxin Impact on the Proteasome in Breast Cancer Cells

In order to identify molecular mechanisms impinged by OU and Digo, which are relevant for both the degradation of the ERα and for proliferation of cells modeling primary and MBC, we performed Affymetrix gene expression profile of MCF-7, Tam Res, and Y537S cells treated for 24 h with either OU or Digo (100 nM). Notably, we chose to treat cells with the same concentration for either CG to avoid any technical bias due to different drug concentration.

Gene expression profile analysis revealed that OU treatment in all BC cells significantly modulated 2226 genes (*q*-value < 0.05) (Appendix A), while Digo treatment modulated 2153 genes (*q*-value < 0.05) in all BC cells (Appendix A), with a significant overlap between the two gene sets, of 1328 genes (*p*-value < 0.0001; Fisher’s test). The substantial overlap between OU- and Digo-regulated genes further suggests a common mechanism of action of the two drugs in BC cells.

Ingenuity pathway analysis (IPA) confirmed common, significantly represented pathways (*q*-value < 0.05) among OU- and Digo-regulated genes, with the top three pathways related to ‘protein ubiquitination’, ‘tRNA charging’, and ‘cholesterol biosynthesis’ (Appendix A). Moreover, IPA upstream regulator analyses for the CGs-modulated genes predicted the common inactivation of a variety of transcription factors (e.g., SREBP2, NRF2, SOX11, TBX2) (Appendix A). Accordingly, Western blotting analysis confirmed that in MCF-7 and in Y537S cells, OU and Digo reduced the protein expression of both SREBP2 and of SREBP1, which is the other human sterol regulatory element-binding protein (Appendix A).

We additionally studied the impact of OU and Digo on the ubiquitination pathway (i.e., ubiquitination cascade and proteasome) in BC cells. Dose response curves were performed to evaluate the total ubiquitinated species in MCF10a, MCF-7, ZR-75-1, Y537S, and Tam Res cells. Figure 2A–L demonstrates that OU and Digo reduce the total content of ubiquitinated species in a dose-dependent manner, depending on the cell type, with a dramatic effect observed between 10^−7^ and 10^−6^ M, with the exception of MCF10a. These data support the notion that these CGs deregulate the proteasome/ubiquitination pathway.

Next, we speculated that the CGs could influence the activity of the ubiquitin-activating enzyme (E_1_), since it has been previously demonstrated that its inhibition leads to a reduction in total ubiquitinated species [32,33]. Therefore, we measured the activity of E_1_ in the presence and absence of OU in in vitro ubiquitination assays [34]. Figure 2M shows that preincubation of E_1_ with different concentrations of OU did not affect its ability to form thioester bonds with Ub on their catalytic cysteine. Thus, OU was not able to inhibit the E_1_ activity in vitro. Despite this evidence, we speculated that the reduction in total ubiquitinated species in the cells could also be dependent on a reduction in the intracellular content of the E_1_ enzyme. Therefore, we measured the levels of UBE1 (i.e., E_1_) in both MCF-7 and ZR-75-1 cells treated with different doses of OU and Digo for 24 h, and observed that the CGs do not change the UBE1 intracellular levels (Figure 2N,N’). Therefore, OU does not affect the activity of the E_1_ enzyme, and neither one of the CGs significantly changes the E_1_ levels.

These data indicate that OU and Digo reduce the total protein ubiquitination levels in a dose-dependent manner in various BC cell lines, but not in non-transformed breast MCF10a cells. This effect is not attributed to either an inhibition of the in vitro activity of the E_1_, nor to a significant modification of the E_1_ intracellular levels.

Nevertheless, as hierarchical clustering analysis returned a large OU- and Digo-dependent downregulation of genes encoding for the proteasome subunits (Figure 3A,B, and Appendix A), we evaluated whether OU and Digo could affect proteasomal function. For that purpose, we initially investigated the effect of the CGs in MCF-7 and ZR-75-1 cells in the presence and in the absence of the proteasome inhibitor Mg-132, in order to evaluate the overall proteasome-degradation functions, using p53, a protein well-known for being targeted by the proteasome for degradation [35].

We observed that treatment with the CGs reduced the total levels of ubiquitinated proteins, as well as the p53 levels in either cell lines, while, as expected, Mg-132 administration to MCF-7 and ZR-75-1 cells effectively increased them (Figure 3C,D). Surprisingly, we additionally observed that both OU and Digo strongly reduced the Mg-132-induced accumulation of both total ubiquitinated species and p53, in both cell lines (Figure 3C,D). Altogether, these results indicate that OU and Digo could alter the homeostasis of the proteasome pathway functions in BC cells.

### 3.3. Ouabain and Digoxin-Dependent Activation of the Proteasome

Prompted by these observations, we hypothesized that the effect of the CGs on the total ubiquitinated species in BC cells could be attributed to their ability to directly affect proteasome activity. To test this hypothesis, we treated MCF10a, ZR-75-1, MCF-7, Y537S, and Tam Res cells with different doses of both OU and Digo for 24 h, and measured all three protease activities of the proteasome (i.e., chymotrypsin-like, caspase-like, and trypsin-like) [27]. In addition, we used the proteasome inhibitor Mg-132 as an internal control for the measurement of proteasome activity [27]. Mg-132 efficiently inhibited all three activities of the proteasome, with an efficacy comparable to previously observed data [27] (Figure 4A’,B’ and Appendix A). Interestingly, we found that both OU and Digo increased, in a dose-dependent and cell type-specific manner, all the proteasome activities in the cells (Figure 4A,B and Appendix A). In contrast, no change in any of the proteasome activities was detected when non-transformed breast MCF10a cells were treated with different doses of OU and Digo (Figure 4A,B and Appendix A). Therefore, these data suggest that OU and Digo could activate the proteasome in BC cells.

Next, we evaluated whether OU could directly activate the proteasome. For this purpose, we detected in vitro all three protease activities of the proteasome, by administering different doses of both OU and Mg-132 to the recombinant commercially available proteasome. Figure 5A demonstrates that in vitro OU increased all proteasome activities in a dose-dependent manner, while, as expected, Mg-132 inhibited them (inset in Figure 5A) [27]. Next, we measured the IC_50_ of Mg-132 for each proteasome activity in the presence of OU, and found that increasing doses of OU augmented the IC_50_ of Mg-132 for chymotrypsin-like, caspase-like and tryspin-like activities of the proteasome. A prominent significant effect was observed, especially for the chymotrypsin-like activity (Figure 5B). Thus, OU in vitro competes with the inhibitory activity of the Mg-132 on the recombinant proteasome.

We next treated both MCF-7 and ZR-75-1 cells with different doses of Mg-132, both in combination with different doses of OU for 24 h. We detected Mg-132-induced increase in total ubiquitinated species, which is a marker for proteasome inhibition in cell lines. Dot blot analyses were performed on cell lysates spotted on nitrocellulose, and subsequently stained with anti-Ub antibody. As expected, Mg-132 increased the amount of total ubiquitinated species in cells in a dose-dependent manner (Figure 5C,D), with a calculated ED_50_ < 0.5 µM, both for MCF-7 and ZR-75-1 cells (Figure 5E). Interestingly, treatment of the two BC cell lines with OU augmented the ED_50_ of Mg-132 with regard to the accumulation of ubiquitinated species, in a dose-dependent manner (Figure 5E). Finally, we validated the above-mentioned observations by performing regular Western blotting analyses in MCF-7 and ZR-75-1 cells, treated with different doses of OU and Digo, both in the presence and absence of Mg-132 (1 µM), for 24 h. As expected, OU and Digo dose-dependently reduced the total ubiquitinated species in both cell lines, while Mg-132 increased them (Figure 5F–I). Remarkably, OU and Digo treatment reduced the Mg-132-dependent accumulation in cellular ubiquitinated proteins as a function of the administrated dose, both in MCF-7 (Figure 5F,G) and in ZR-75-1 (Figure 5H,I) cells. These data demonstrate that OU and Digo enhance proteasome activity, and directly compete with Mg-132 for the proteasome activity in the cells.

### 3.4. Ouabain and Digoxin Impact on E2:ERα Signaling

ERα binding assay was performed with different doses of OU, Digo and E2, in order to test whether OU and Digo could directly bind ERα in vitro.

Only E2 (Figure 6A) was able to displace the fluorescent E2, used as a tracer for the recombinant purified ERα in vitro, with a measured IC_50_ (i.e., K_d_) of approximately 2.9 nM, as previously reported [36]. Next, the ERα mRNA levels in MCF-7 cells, treated with OU and Digo for 24 h were measured, and found not to be modified following CGs administration (Figure 6B). These results indicate that OU and Digo do not bind to ERα in vitro, and do not increase ERα mRNA levels.

Next, on the basis of OU and Digo’s observed reduction of the ERα content in BC cells, we tested their effect on ERα function, by evaluating E2 signaling to cell proliferation in MCF-7 and ZR-75-1 cells. ERα is a ligand-induced transcription factor that regulates the expression of diverse genes, both those containing and those not containing the estrogen response element (ERE) in their promoters [2]. Therefore, the effect of CGs on E2-induced ERα transcriptional activity was measured by evaluating the expression of ERE- and non-ERE-containing genes upon E2 treatment.

As expected, the E2-evoked increase in the intracellular content of ERE- and non-ERE-containing genes (i.e., pS2/TFF1, cathepsin D–CatD, cyclin D1–cyc D1, and Bcl-2) was strongly prevented by both OU and Digo pretreatments, in both MCF-7 (Figure 7A,C) and ZR-75-1 (Figure 7B,D) cells. As expected, ICI administration to BC cells also prevented the E2-induced accumulation of the above-mentioned proteins, and both ICI, OU, and Digo treatment reduced ERα intracellular levels (Figure 7A,B) in MCF-7 cells. Notably, E2 was able to induce receptor degradation, both in the absence and in the presence of CG administration to BC cells (Figure 7A–D). These data suggest that CGs reduce E2-induced receptor transcriptional activity.

In order to substantiate these findings, we directly compared the effect of ICI, OU, and Digo, on two E2-regulated genes (i.e., pS2/TFF1, and CatD), in both MCF-7 and Y537S cells [37]. We chose Y537S cells, since their mutated ERα is a transcriptional hyperactive receptor variant, which assumes a constitutively active agonist structural conformation, identical to that of the wild type (wt) receptor bound to the cognate hormone E2 [15,38]. As shown in Figure 7E,F, pS2/TFF1 and CatD expression was upregulated in Y537S with respect to MCF-7 cells, as previously reported [37]. Interestingly, while treatment of MCF-7 cells with ICI, OU, and Digo barely affected the basal expression levels of the above-indicated genes, their administration into Y537S cells reduced the intracellular concentration of both pS2/TFF1 and CatD (Figure 7E,F). Notably, the observed effect of both OU and Digo in reducing the abundance of pS2/TFF1 and CatD appeared to be superior to that achieved by ICI administration (Figure 7E,F).

The E2-dependent activation of ERα in BC cells results in cell cycle progression and cell proliferation [2]. In turn, we studied the effect of CGs on the ability of E2 to induce cell proliferation in MCF-7 and ZR-75-1 cells. As expected, E2 induced an increase in cell number after 48 h of treatment (Figure 7G,H). Pretreatment of BC cell lines with either OU or Digo prevented both the basal and the E2-induced increase in cell number, in both MCF-7 (Figure 7G), and in ZR-75-1 (Figure 7H) cells.

Altogether, these data indicate that treatment of BC cells with CGs prevents E2-induced ERα-dependent transcriptional activity, as well as the ability of E2 to induce cell proliferation.

### 3.5. Ouabain and Digoxin-Dependent Proteasome Activation Precedes the Antiproliferative Effects

The data we obtained on the effect of OU and Digo on BC cells indicate that they reduce the levels of ERα within 24 h, leading to E2-insensitivity, block the cell cycle in the G2 phase, hyperactivate the proteasome, and induce BC cell death.

Therefore, we sought to determine the chronological succession of the observed effects. We treated the ZR-75-1 cells with OU and Digo for 8 h, and measured ERα and general protein ubiquitination levels and proteasome activity, and performed cell cycle analysis.

Under these conditions, both OU and Digo were able to reduce the cellular levels of ERα and of total ubiquitinated proteins (Figure 8A,D). In addition, after 8 h, while the CGs increased proteasome activity in a dose-dependent manner, and the Mg-132 efficiently inhibited them (Figure 8B,E and relative insets), they did not induce any significant variations in the cell cycle phases (Figure 8C,F).

Therefore, these data suggest that the OU- and Digo-dependent proteasome activation occurs prior to cell cycle block in G2.

### 3.6. Preclinical Evaluation of Ouabain and Digoxin as Novel Drugs for Treatment of Primary and Metastatic BC

Since OU and Digo independently exhibit antiproliferative activities in cells modeling primary and metastatic BC, we next evaluated whether they could display synergistic effects with endocrine therapy drugs already used for treatment of primary and metastatic BC.

Therefore, we first treated MCF-7 and ZR-75-1 cells with increasing concentrations of 4OH-Tam (10–100,000 nM), OU and Digo (10–100 nM). Combination analyses of both 4OH-Tam with OU or Digo significantly reduced the number of both MCF-7 and ZR-75-1 cells (i.e., cellular models of primary BC) with respect to both untreated, 4OH-Tam-, and OU- and Digo-treated cells (Figure 9A,B). Contrarily, combination analyses in Y537S cells (i.e., a cellular model of MBC) with 4OH-Tam or ICI, and the novel SERDs AZD9496 and GDC-9545 [2,10,39] did not exhibit any combinatorial effects (data not shown). These data indicate that OU and Digo demonstrate synergism with known anti-BC drugs in preventing cell proliferation, and further suggest that they could be used as an adjuvant drug in the treatment of primary, rather than MBC.

Next, we studied the antiproliferative effects of OU and Digo in MCF-7 and Y537S tumor cell spheroids, also using ICI as internal control. Tumor spheroids were counted at time 0 (i.e., before drug administration), and at the end of the treatment (i.e., 7 days). Both MCF-7 and Y537S cell spheroids grew within the experimental window (i.e., 7 days), and ICI was able to prevent MCF-7, but not Y537S spheroid growth (Figure 9C,C’). On the contrary, both OU and Digo reduced the number of MCF-7 and Y537S spheroids (Figure 9C,C’). Therefore, these data indicate that OU and Digo maintain their antiproliferative activity also in a 3D environment.

Prompted by these results, we additionally studied the ability of the CGs to demonstrate anticancer effects in an in vivo model of a breast tumor. For this purpose, we generated xenograft of Y537S cells in a Zebrafish environment, which is recognized as a reliable cancer model for testing the efficacy of anti-BC drugs [40,41]. Figure 9D shows that treatment of Y537S cell xenograft with OU and Digo resulted in a statistically significant reduction in the primary tumor area after 4 days of treatment, in comparison with untreated control xenografts. Therefore, the CGs reduce tumor mass also in the context of a living organism.

Finally, we investigated whether the OU- and Digo-regulated genes derived by Affymetrix profiling could identify putative OU or Digo responsive patients with 4OH-Tam-resistant human BC. For this purpose, we downloaded gene expression data of a cohort of 177 TCGA-BRCA patients treated with 4OH-Tam, and the relevant information about treatment response. BC samples were separated into hormone-therapy sensitive (i.e., with a complete response, CR, N = 163; RECIST criteria), and hormone-therapy resistant (i.e., with progressive disease, PD, N = 6; RECIST criteria) samples. We then performed Gene Set Enrichment Analysis (GSEA) of the cohort of BC samples, using the common top 100 upregulated and top 100 downregulated genes by OU or Digo (Appendix A). Strikingly, we found a significant enrichment (NES > 1.5; FDR < 5%) of the ‘top 100 downregulated genes’ among the core of upregulated genes in PD patients (Figure 9E). Similar results were obtained in hormone-therapy sensitive (CR) and in hormone-therapy resistant samples, but not for the stable disease samples (SD, N = 8; RECIST criteria) (Figure 9F). These results suggest that the signature of the overexpressed genes in hormone-resistant patients could be reverted by treatments with OU and Digo.

Overall, these data demonstrate that OU and Digo exert synergistic antiproliferative activity with classic ‘anti-estrogen’ drugs, work in 3D models of breast tumors, and stratify different classes of patients as potentially responsive to these drugs.

**Figure 9 cancers-12-03840-f009:**
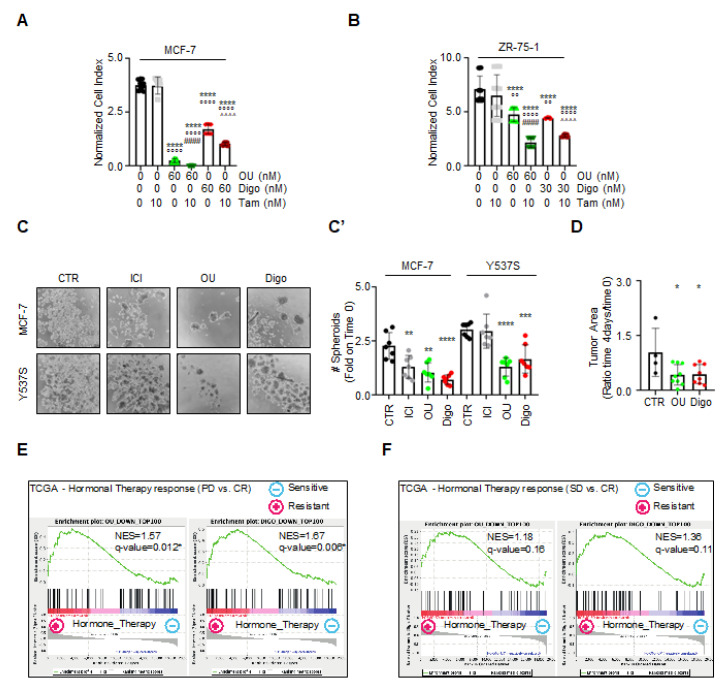
The effect of ouabain and digoxin in preclinical models of breast cancer. Normalized cell index (i.e., cell number, as described in Section 2.3) measured in MCF-7 (**A**), and ZR-75-1 (**B**) for 5 days, with the combination of the indicated drugs. Significant differences with respect to the 0, 0 sample were determined by unpaired two-tailed Student’s *t*-test: **** *p*  <  0.0001. Significant differences with respect to the 4OH-Tamoxifen (4OH-TAM) sample were determined by unpaired two-tailed Student’s *t*-test: °°°° *p*  <  0.0001; °° *p*  <  0.01. Significant differences with respect to the ouabain (OU) sample were determined by unpaired two-tailed Student’s *t*-test: #### *p*  <  0.0001. Significant differences with respect to the digoxin (Digo) sample were determined by unpaired two-tailed Student’s *t*-test: ^^^^ *p*  <  0.0001. Images (**C**) and quantitation (**C’**) of tumor spheroids generated in MCF-7 and Y537S ERα-expressing MCF-7 (Y537S), treated at time 0 with ICI182,780 (ICI–100 nM), ouabain (OU–100 nM), digoxin (Digo–100 nM), or left untreated (CTR), for 7 days. Experiments were performed twice, in quadruplicate. Significant differences with respect to the CTR sample were determined by unpaired two-tailed Student’s *t*-test: **** *p*  <  0.0001; *** *p* < 0.001; ** *p* < 0.01. (**D**) The effect of DMSO (four fishes were used), OU (10 µM) (nine fishes were used), and Digo (1.5 µM) (eight fishes were used) on the tumor area in Zebrafish xenografts of Y537S ERα-expressing MCF-7 (Y537S) cells treated for 4 days. Significant differences with respect to the CTR sample were determined by unpaired two-tailed Student’s *t*-test: * *p*  <  0.05. Details are given in Section 2.12. These experiments were performed by ZeClincs. The related final report is available upon request. GSEA analysis of ouabain (OU) and digoxin (Digo) gene sets (i.e., the top 100 downregulated genes) in hormone-resistant patients (PD) vs. complete responders (CR) (**E**), and vs. stable disease patients (SD) (**F**) of the TCGA-BRCA cohort. NES, normalized enriched score; *q*-value, significance of enrichment based on 1000 random gene permutations.

## 4. Discussion

ERα targeting through specific inhibitor ligands remains paramount in the treatment of primary and metastatic BC [1,2,7,8,9,10,42]. However, we and others have also demonstrated that the sole modification of ERα intracellular concentration in BC cells is sufficient to reduce or block cancer cell proliferation, even if ERα is not directly targeted by the drug [2,43,44]. In this respect, we found OU and Digo as FDA-approved drugs that reduce ERα intracellular content and cell proliferation in ductal carcinoma cells (MCF-7 cells), in a previous drug screen [18]. Here, we characterized the effects of these CGs in BC cells, in order to pave the way for their application as an alternative therapeutic strategy for the treatment of primary and/or metastatic BC, either separately, or in combination with classic endocrine therapy drugs.

By using different cellular models mimicking primary and metastatic BCs, we found that OU and Digo induce ERα degradation without binding to ERα, prevent E2:ERα-mediated transcriptional activity, and block E2-induced cell proliferation, inducing apoptosis by blocking the cell cycle in the G2 phase, and are small-molecule activators of the proteasome. Time-dependent analysis also suggests that the CGs induce proteasome activation, ERα protein reduction, E2 insensitivity, cell cycle blockade, and apoptosis (Figure 10). Remarkably, the CGs-induced effects are at least partially independent of the inhibition of Na/K ATPase, since they occur at concentrations below (or comparable to) the measured IC_50_ of the Na/K pump for each cell line. Moreover, they also occur in ZR-75-1 cells, where OU and Digo do not inhibit Na/K ATPase activity [45,46].

Over the last decade, FDA-approved CGs drugs were demonstrated as effective anticancer drugs [20]. Accordingly, our data confirm the antiproliferative activity of OU and Digo in primary and metastatic BC cell lines. We observed that they block the cell cycle in the G2 phase, and induce apoptosis. OU and Digo work as antiproliferative agents in cellular models of MBC resistant to 4OH-Tam treatment. In addition to ERα degradation, our BC transcriptome analyses provide further evidence to elucidate the antiproliferative effects of these CGs in MBC cells. Indeed, both OU and Digo preferentially reduce the genes in the cholesterol metabolism pathway, by reducing the levels of the transcription factors SREBP1 and SREBP2. Interestingly, it is increasingly recognized that MBC cells appear to be dependent on cholesterol metabolism [47]. Thus, OU and Digo could revert metabolic reprogramming by switching off cholesterol biosynthesis in MBC, causing cell death.

Moreover, IPA analysis shows that OU and Digo could reduce the activity of many transcription factors in which overexpression is required for sustained proliferation and cancer progression (e.g., NRF2, SOX11, TBX2) [48,49,50,51]. This implies that these CGs could generally impact the transcriptional networks required for sustained cancer progression. In this respect, it is important to point out that the differences in the sensitivity to CGs we observed in Tam Res cells with respect to ERα degradation and cell proliferation could indicate that while on one hand these cells are more resistant to the degradation of the receptor, on the other hand they could be more susceptible to the degradation of unknown critical proteins that regulate cell proliferation.

OU and Digo induce ERα degradation in all tested cell lines, including the one expressing the ERα Y537S mutated variant, which is resistant to SERD-induced degradation [2]. ERα degradation is not initiated by the direct binding of OU and Digo to the receptor, and not even by an OU- and Digo-dependent reduction in ERα mRNA levels. However, Affymetrix analyses and subsequent experiments demonstrated that OU and Digo are small molecule activators of the proteasome. On this basis, we propose that the CGs-induced proteasome hyperactivity most likely results in the observed CGs-induced ERα degradation.

Measurement of the activities of the proteasome in various cell lines demonstrates that OU and Digo differentially affect the chymotrypsin-like, caspase-like, and trypsin-like activity of the proteasome in a cell-type-dependent manner. Importantly, in our cell-based studies, we additionally report a specificity of OU and Digo in activating the proteasome in BC cells, while the proteasome activity in non-transformed breast cells (i.e., MCF10a) appears barely affected. This is possibly because different proteasome subunit compositions between non-transformed and tumor cell lines could exist. Unfortunately, we were not able to study in vitro the ability of Digo to affect proteasome activity, because its solvent (i.e., DMSO) interferes with in vitro measurements. Nonetheless, in vitro OU increases all three activities of the recombinant commercially available proteasome, and competes with the proteasome inhibitor Mg-132 for the modulation of the proteasome activity. Cellular studies further confirmed these results, and indicated that Digo also exerts such effects on two different BC cell lines. At the present, although the structural biochemical mechanism(s) underlying OU- and Digo-dependent proteasome activation are unknown, in vitro data for OU suggest that these CGs could directly bind to some proteasome subunits, the identification of which would be made possible through bioinformatic analyses (i.e., docking studies).

Competition studies, both in vitro and in cell lines, strongly indicate that OU and Digo can work as allosteric activators of the proteasome, as they interfere with the Mg-132 inhibitor function in the proteasome. In this respect, since Mg-132 has a higher affinity for the chymotrypsin-like activity of the proteasome [27], and OU appears to compete with Mg-132 specifically for this enzymatic function, it is tempting to speculate that OU could bind to the region of the proteasome directly required for chymotrypsin-like activity. Accordingly, the fact that within 24 h, OU and Digo reduce the expression of the proteasome subunit genes, while within 8 h they induce the activation of the proteasome enzymatic activities, leads us to speculate that cells activate a rapid compensatory mechanism to buffer the CGs-dependent proteasome activation. Interestingly, the ability of OU to activate the proteasome can also explain some of the previously reported effects of this CG on specific components of the ubiquitination pathway [52].

To date, no small molecules that directly activate the proteasome in the nanomolar range are available [53], with the exception of the herein identified OU and Digo. Proteasome activation is being recognized as a therapeutic strategy to treat proteotoxic disorders [54]. Thus, present discoveries render these FDA-approved CGs appealing for the treatment of other pathologies in addition to BC, including some neurological disorders, where the cytosolic accumulation of undegradable proteins is a pathogenetic cause of disease. Accordingly, a similar proteasome activator-based approach has been previously proposed for Huntington’s disease [55].

Nonetheless, the present data also suggest caution should be applied in the use of these CGs in normal cells, and disclose the necessity to accurately choose concentrations of OU and Digo, and control their effects on the proteasome during the developmental phase of a CGs-based pharmacological treatment. Indeed, although we demonstrated that OU and Digo do not activate the proteasome in a non-transformed cell line, available data show that high proteasome activity relates to cancer development [56,57]. However, at the same time, our data strongly indicate that the OU and Digo ability to activate the proteasome is prospective for the treatment of ERα-expressing BCs. In this respect, BC patients who were treated with CGs (i.e., digitalis protocol) before being diagnosed with cancer have a significantly lower mortality rate than the patients that did not previously undergo digitalis [58]. Moreover, presently, administration of digitalis drugs is not considered as treatment for women diagnosed with BC, and no clinical studies exist to demonstrate the efficacy of digitalis administration in women with BC. Thus, our results offer a new treatment option for ERα+ BC patients, especially because digitalis drugs affect a cellular pathway (i.e., proteasome) that controls ERα abundance, and consequently its activity in BC cells. Such a novel strategy aligns with the concept that in cancers, both strong inhibition and hyperactivation of a specific target are valuable options for therapeutic treatment of the disease [59].

In addition, in the perspective of ERα+ breast tumors, it is important to note that OU and Digo block E2 signaling for cell proliferation. Indeed, we observed that they prevent the accumulation of those proteins, whose genes are under the control of the E2-activated ERα. More importantly, both OU and Digo are effective in reducing the transcriptional hyperactive E2-mimetic activity of the ERα Y537S mutant. Therefore, OU and Digo strongly reduce ERα transcriptional activity and, in turn, their ability to proliferate in response to E2. Notably, the blockade of E2:ERα transcriptional signaling, and consequently cell proliferation observed for these CGs is similar to the effects elicited by aromatase inhibitors and 4OH-Tam [1], suggesting that these drugs could be useful for the treatment of primary BCs. Notably, OU and Digo preferentially synergize with classic endocrine therapy drugs (e.g., 4OH-Tam) in primary BC cells (see below), indicating that they could be used as adjuvant drugs in the context of a primary breast tumor.

The OU and Digo cardiotoxic effects [20] remain an issue when considering their clinical potentials as anticancer drugs. In this perspective, our analyses have shown that the activation of the proteasome, the reduction in ERα protein levels, and blockade in cell proliferation occur within the nanomolar range, which is compatible with the plasma concentrations achieved in patients treated with FDA-approved doses of the drugs [60]. Nonetheless, our coadministration studies with classic or novel endocrine therapy drugs (i.e., 4OH-Tam, ICI, AZD. and GDC) in cell lines exhibiting primary (i.e., MCF-7 and ZR-75-1 cells) and metastatic BCs (i.e., Y537S cells) reveal that OU and Digo can synergize with endocrine therapy drugs only in primary BC cells. In this respect, it is important to note that the doses of 4OH-Tam, OU, and Digo can be efficiently scaled down in the low nanomolar range (i.e., 30–60 nM) to determine a synergic antiproliferative activity. Moreover, OU and Digo maintain their antiproliferative effects at nanomolar concentrations also in a 3D culture model of BC (tumor spheroids), and display antitumor effects in Zebrafish xenografts of MBC cells (i.e., Y537S).

Importantly, we also demonstrated that OU and Digo downregulate genes which are among the most overexpressed in patients with BC resistant to 4OH-Tam treatment. Although these observations warrant further investigation to explore the efficacy of OU and Digo as an alternative therapeutic strategy for 4OH-Tam-resistant patients, they open the possibility to explore the effect of these drugs as potential additional treatments for MBC, for which a standard therapeutic protocol is still not available. Thus, overall, our present preclinical investigations suggest that OU and Digo could work as effective antiproliferative agents in an organized tumor environment, at nanomolar doses.

## 5. Conclusions

Our discoveries demonstrate that OU and Digo could work in the cellular level as ‘anti-estrogen’-like compounds, and provide the basis for further clinical research to evaluate the possibility to reallocate them as drugs for treating primary and metastatic BCs that became insensitive to endocrine therapy drugs.

## Figures and Tables

**Figure 1 cancers-12-03840-f001:**
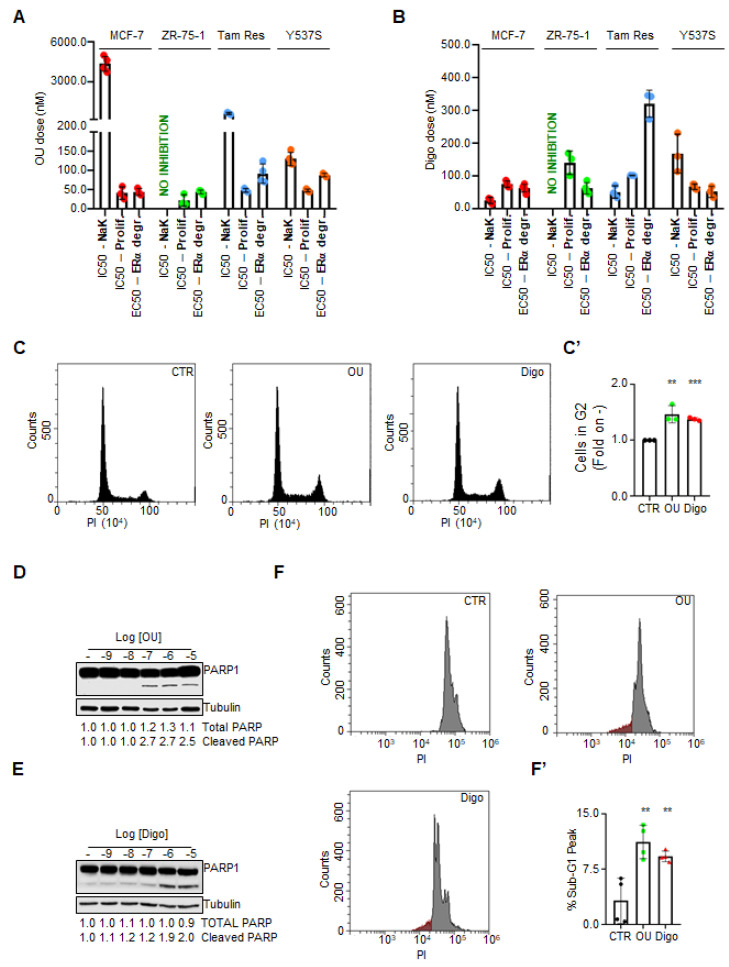
Effective concentrations analysis of the ouabain and digoxin effects on ERα expression, cell proliferation, and Na/K ATPase activity in breast cancer cells. (**A**) Effective dose 50 (ED_50_–nM) of ouabain (OU) for reduction in ERα intracellular levels in the indicated cell lines, calculated after 24 h of drug administration to each cell line; inhibitor concentration 50 (IC_50_–nM) of OU for cell proliferation in the indicated cell lines, calculated after 5 days of treatment for each cell line; IC_50_ (nM) for OU for Na/K ATPase activity in the indicated cell lines, calculated after 24 h of treatment for each cell line. (**B**) ED_50_ (nM) of digoxin (Digo) for the reduction in ERα intracellular levels in the indicated cell lines, calculated after 24 h of drug administration to each cell line; IC_50_ (nM) of Digo for cell proliferation in the indicated cell lines, calculated after 5 days of treatment for each cell line, and IC_50_ (nM) of Digo for Na/K ATPase activity in the indicated cell lines, calculated after 24 h of treatment for each cell line. Cell cycle analysis (**C**) and relative quantitation of the G2 phase (**C’**) in ZR-75-1, treated for 24 h with either ouabain (OU—100 nM), or digoxin (Digo—1000 nM), and for untreated cells (CTR). Significant differences with respect to the CTR sample were obtained by unpaired two-tailed Student’s *t*-test: *** *p*  <  0.001; ** *p*  <  0.01. Western blotting analyses of PARP1 expression levels in ZR-75-1 cells, treated for 24 h with ouabain (OU) (**D**), and digoxin (Digo) (**E**) at the indicated doses. The loading control was performed by evaluating tubulin expression in the same filter. Experiments were performed twice in duplicates. Cell cycle analysis (**F**) and relative quantitation (**F’**) for the evaluation of the sub-G1 phase in ZR-75-1, treated for 72 h with ouabain (OU—100 nM), or digoxin (Digo—1000 nM), or untreated (CTR). Experiments were performed four times. Significant differences with respect to the CTR sample were obtained by unpaired two-tailed Student’s *t*-test: ** *p*  <  0.01.

**Figure 2 cancers-12-03840-f002:**
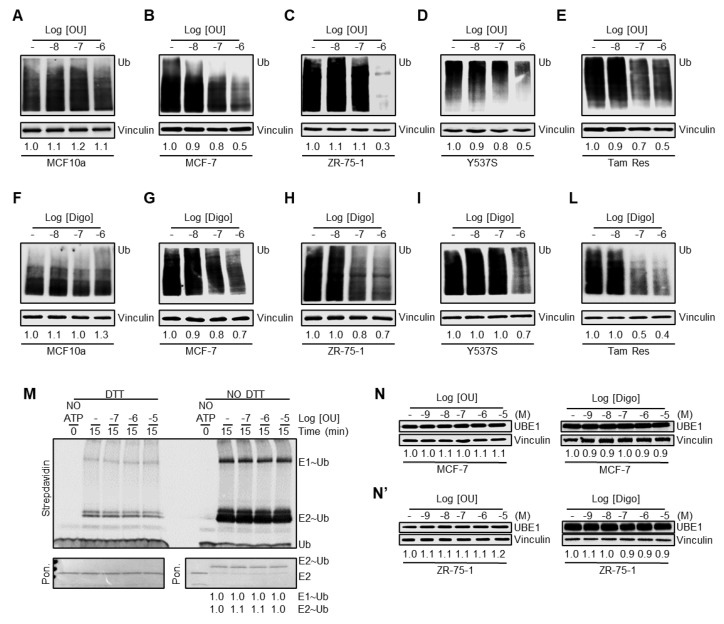
The effect of ouabain and digoxin on the ubiquitin-proteasome pathway. Western blotting analyses of ubiquitin (Ub) expression in the indicated cell lines, treated for 24 h with the indicated doses of ouabain (OU) (**A**–**E**), and digoxin (Digo) (**F**–**L**). Experiments were repeated three times. (**M**) Western blotting analyses of streptavidin-detected ubiquitin-activating (E_1_) and ubiquitin-conjugating (E_2_) enzyme-dependent biotinylated Ub (Ub) thioester formation, in the presence of the indicated doses of ouabain (OU), for 15 min. The transfer of Ub to E_1_ (E_1_~Ub) and E_2_ (E_2_~Ub) was monitored by quenching the reaction by addition of Laemmli buffer, with and without the reducing agent (100 mM DTT). Right panel (without DTT-NO DTT) indicates thioester loaded E_1_ (E_1_~Ub) and E_2_ (E_2_~Ub). Left panel (DTT-resistant bands-DTT) represents self-ubiquitinated enzymes. Ponceau (Pon.) staining of the membrane after transfer was used to show equal loading of the enzymes in all samples, both in the presence and in the absence of ATP (NO ATP-0). Experiments were performed twice. Western blotting analyses of the ubiquitin-activating enzyme (UBE1) expression in MCF-7 (**N**) and ZR-75-1(**N’**) cells, treated for 24 h with the indicated doses of ouabain (OU), digoxin (Digo), and untreated (-). Experiments were performed twice.

**Figure 3 cancers-12-03840-f003:**
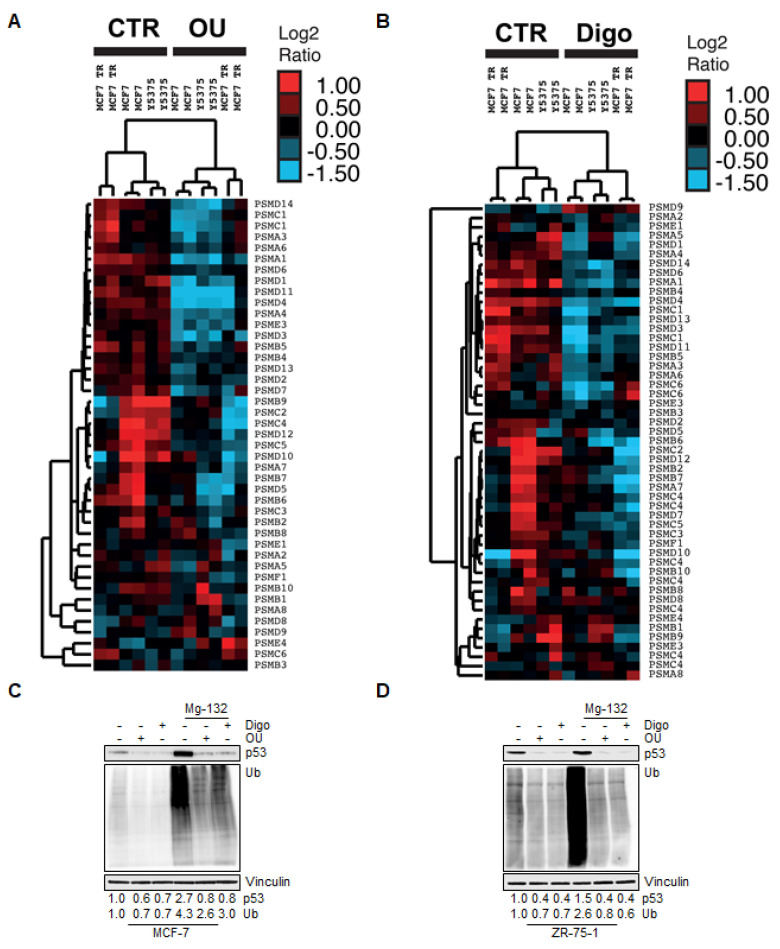
The effects of proteasome genes regulated by ouabain and digoxin on breast cancer cells. Hierarchical clustering analysis of gene expression profiles, relative to 20S proteasome-related gene set, in cells treated for 24 h with OU (**A**) and Digo (**B**). Experiments were performed in duplicates. Western blotting analyses of p53, and ubiquitin (Ub) expression in MCF-7 (**C**) and ZR-75-1 (**D**) cells treated for 24 h with ouabain (OU–100 nM) or digoxin (Digo–1000 nM), both in the presence and in the absence (CTR) of the administration of the proteasome inhibitor Mg-132 (0.5 µM). All experiments were performed twice in duplicates. The loading control was performed by evaluating vinculin expression in the same filter.

**Figure 4 cancers-12-03840-f004:**
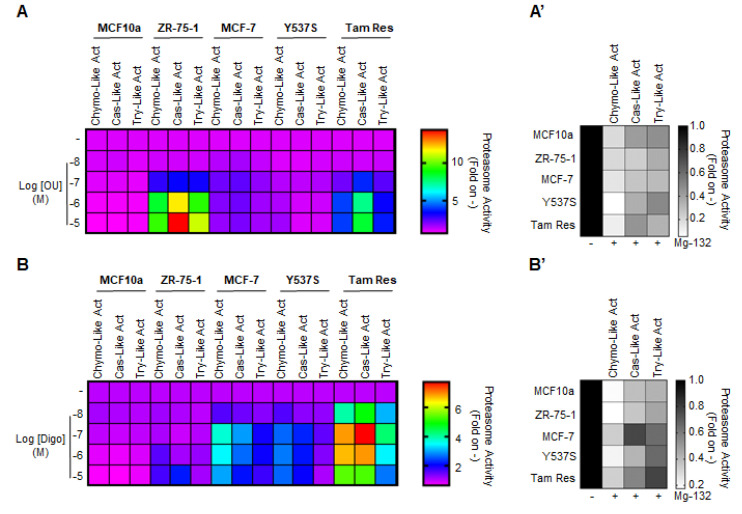
The effect of ouabain and digoxin on proteasome activity in cell lines. Evaluation of the proteasome activities (i.e., chymotrypsin-like, caspase-like, and trypsin-like) in the indicated cell lines, treated for 24 h with the indicated doses of ouabain (OU) (**A**) and digoxin (Digo) (**B**). (**A’**,**B’**) Effect of the proteasome inhibitor Mg-132 (1 µM) administered to the tested cells for 24 h on all three proteasome activities (i.e., chymotrypsin-like, caspase-like, and trypsin-like). All experiments were performed three times, in triplicates. Heatmaps present the mean values. The original data are provided in the source data file, and the graphs are depicted in Appendix A.

**Figure 5 cancers-12-03840-f005:**
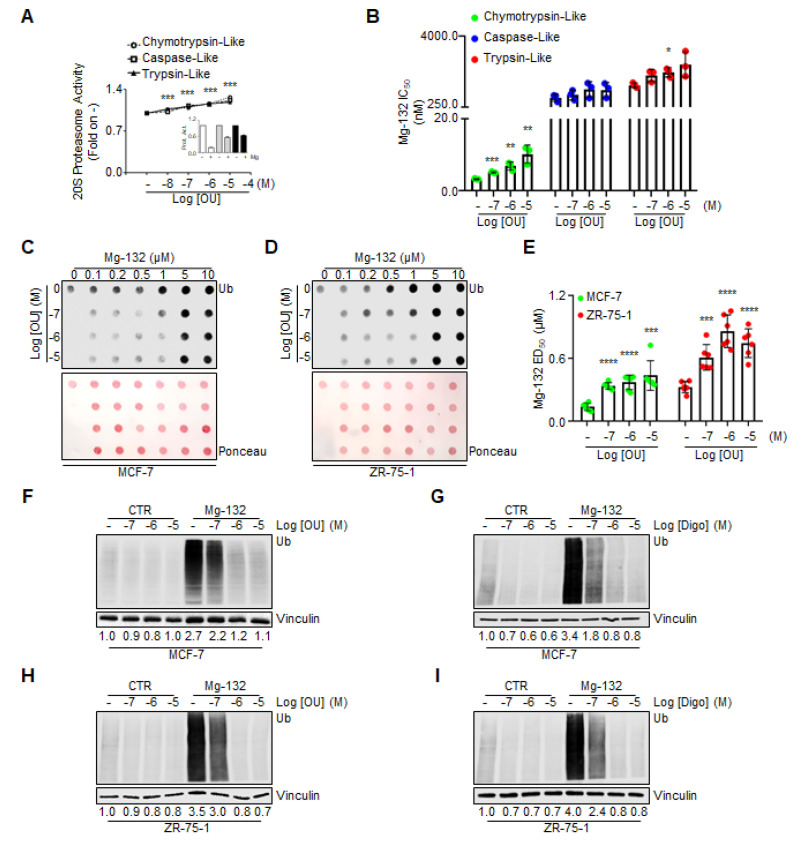
The effect of ouabain and digoxin on proteasome activity in vitro, and in cell lines. (**A**) In vitro evaluation of the proteasome activities (i.e., chymotrypsin-like, caspase-like, and trypsin-like), using recombinant commercially available 20S proteasome, and different doses of ouabain (OU). The inset in panel A indicates the effect of the proteasome inhibitor Mg-132 (1 µM) on all three proteasome activities (i.e., chymotrypsin-like, caspase-like, and trypsin-like). All experiments were performed three times, in triplicates. Significant differences with respect to the—sample were determined by unpaired two-tailed Student’s *t*-test: *** *p*  <  0.001 (for all three activities). (**B**) In vitro evaluation of the inhibitor concentration 50 (IC_50_–nM) of the proteasome inhibitor Mg-132 for each proteasome activity (i.e., chymotrypsin-like, caspase-like, and trypsin-like), using recombinant commercially available 20S proteasome, both in the presence and in the absence of different doses of ouabain (OU). All experiments were performed three times, in triplicates. Significant differences with respect to the-sample were determined by unpaired two-tailed Student’s *t*-test: *** *p*  <  0.001; ** *p*  <  0.01; * *p*  <  0.05. Dot blot of ubiquitin expression (Ub) in MCF-7 (**C**) and ZR-75-1 (**D**) cells treated with the indicated doses of both ouabain (OU) and the proteasome inhibitor Mg-132, for 24 h. (**E**) Quantitation of the effective dose 50 (ED_50_–µM) of the proteasome inhibitor Mg-132 for ubiquitin (Ub) expression in MCF-7 and ZR-75-1 cells, treated with the indicated doses of ouabain (OU). The experiments were performed twice, in triplicates. Significant differences with respect to the-sample were determined by unpaired two-tailed Student’s *t*-test: **** *p*  <  0.0001; *** *p*  <  0.001. Western blotting analyses of ubiquitin (Ub) expression in MCF-7 and ZR-75-1, treated for 24 h with the indicated doses of ouabain (OU) (**F**,**G**), and digoxin (Digo) (**H**,**I**), both in the presence and absence of the proteasome inhibitor Mg-132 (0.5 µM). Experiments were repeated three times. The loading control was performed by evaluating vinculin expression in the same filter.

**Figure 6 cancers-12-03840-f006:**
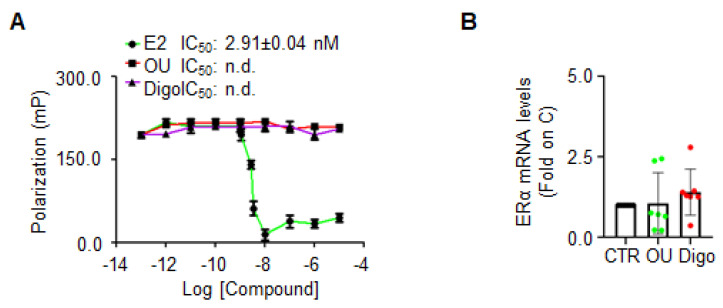
Impact of ouabain and digoxin on ERα binding and mRNA expression. (**A**) In vitro ERα competitive binding assays for ouabain (OU—red), digoxin (Digo—purple), and 17β-estradiol (E2—green) performed at different doses, using a florescent E2 as a tracer. Inhibitor concentration 50 (IC_50_–nM) is indicated in the panel, for each compound. The experiment was performed in quintuplicates. (**B**) Real-time qPCR analysis of ERα mRNA levels in MCF-7 cells, treated for 24 h with ouabain (OU—100 nM) and digoxin (Digo—100 nM). The experiment was repeated seven times.

**Figure 7 cancers-12-03840-f007:**
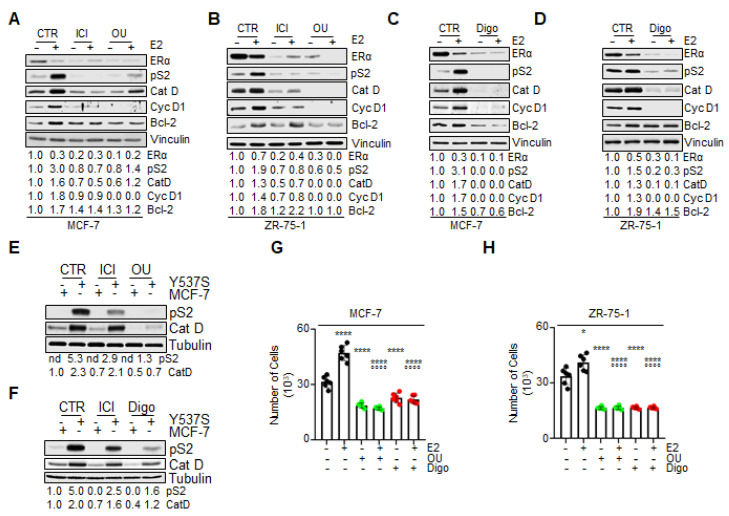
The impact of ouabain and digoxin on E2:ERα signaling in breast cancer cells. Western blotting of ERα, presenilin-2 (pS2), cathepsin D (Cat D), cyclin D1 (Cyc D1), and Bcl-2 expression in MCF-7 (**A**), and ZR-75-1 (**B**) cells, treated, or untreated (CTR) with ouabain (OU–100 nM) or ICI182,780 (ICI–100 nM), in the presence of 17β-estradiol (E2 1 nM), for 24 h. The loading control was performed by evaluating vinculin expression in each filter. All experiments were performed in triplicates. Western blotting of ERα, presenilin-2 (pS2), cathepsin D (Cat D), cyclin D1 (Cyc D1), and Bcl-2 expression in MCF-7 (**C**), and in ZR-75-1 (**D**) cells treated, and untreated (CTR) with digoxin (Digo—1000 nM), in the presence of 17β-estradiol (E2 1 nM), for 24 h. The loading control was performed by evaluating vinculin expression in each filter. All experiments were performed in triplicates. Western blotting of presenilin-2 (pS2), cathepsin D (Cat D), and caveolin-1 (Cav-1) in MCF-7 (**E**) and Y537S ERα-expressing MCF-7 (Y537S) (**F**) cells, treated, and untreated (CTR) with ouabain (OU—100 nM), digoxin (Digo—1000 nM), or ICI182,780 (ICI—100 nM), for 24 h. The loading control was performed by evaluating tubulin expression in each filter. All experiments were performed twice. Number of MCF-7 (**G**) and ZR-75-1 (**H**) cells, treated, and untreated (-) with ouabain (OU—100 nM), or with digoxin (Digo—1000 nM) in the presence of 17β-estradiol (E2 1 nM), for 48 h. Experiments were performed twice, in triplicates. Significant differences with respect to the-sample were determined by unpaired two-tailed Student’s *t*-test: **** *p*  <  0.0001; * *p*  <  0.05. Significant differences with respect to the 17β-estradiol (E2) sample were determined by unpaired two-tailed Student’s *t*-test: °°°° *p*  <  0.0001.

**Figure 8 cancers-12-03840-f008:**
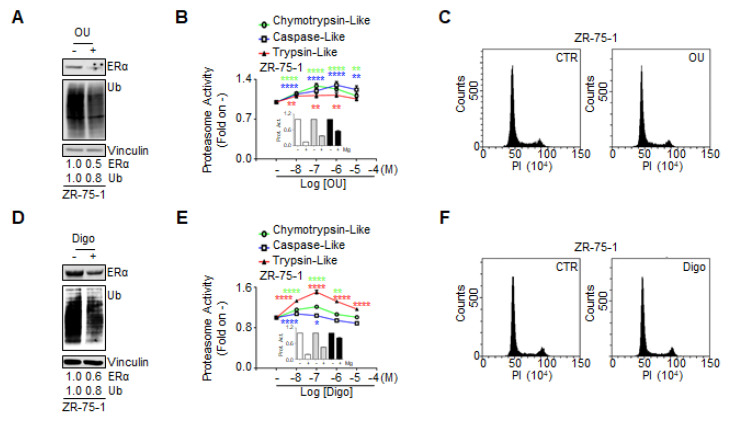
The short-term effect of ouabain and digoxin on ERα expression, proteasome activity, and cell cycle in ZR-75-1 cells. Western blotting analyses of ERα and ubiquitin (Ub) expression in ZR-75-1 cells, treated for 8 h with ouabain (OU—100 nM) (**A**), and digoxin (Digo—1000 nM) (**D**). The loading control was performed by evaluating vinculin expression in the same filter. Experiments were repeated three times. Evaluation of the proteasome activities (i.e., chymotrypsin-like, caspase-like, and trypsin-like) in ZR-75-1 cells, treated for 8 h with different doses of ouabain (**B**) and digoxin (**E**). Insets in panels B–E indicate the effect on all three proteasome activities (i.e., chymotrypsin-like, caspase-like, and trypsin-like) of the proteasome inhibitor Mg-132 (1 µM), administered to ZR-75-1 cells for 8 h. All experiments were performed twice, in triplicates. Significant differences with respect to the—sample were determined by unpaired two-tailed Student’s *t*-test: **** *p*  <  0.0001; ** *p*  <  0.01; * *p*  <  0.05. Cell cycle analyses of ZR-75-1 cells treated for 8 h with ouabain (OU—100 nM) (**C**), and digoxin (Digo—1000 nM) (**F**). Experiments were repeated three times.

**Figure 10 cancers-12-03840-f010:**
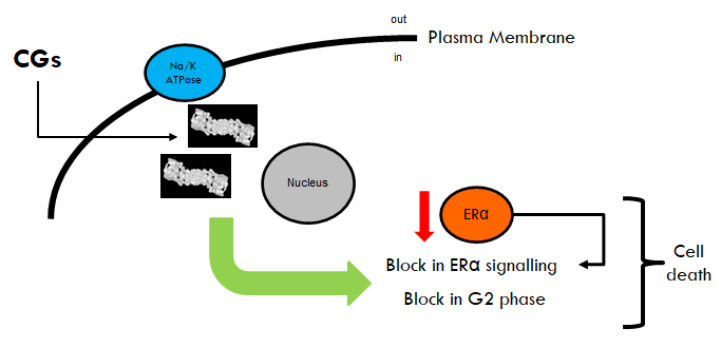
Proposed model for ouabain and digoxin functions in BC cells. Schematic description of the proposed mechanism of ouabain and digoxin (i.e., cardiac glycosides-CGs) in breast cancer cells. A detailed description of the various steps is provided in the main text.

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
