# Peer review of "Ouabain and Digoxin Activate the Proteasome and the Degradation of the ERα in Cells Modeling Primary and Metastatic Breast Cancer"

_cancers, 2020, doi:10.3390/cancers12123840_

Round 1
Reviewer 1 Report
Review of the manuscript untitled:
“Ouabain and digoxin activate the proteasome and degrade the ERα in primary and metastatic breast cancer”
By using a library of FDA approved drugs, the authors previously identified several hits able to down regulate the level of ER . In this manuscript the authors focused on 2 cardiac glycosides : ouabain and digoxin. They showed that these drugs prevent cell proliferation and trigger apoptosis of several BC cell lines. Moreover, they confirmed their results in fish xenografts. Mechanistically, the authors showed that ouabain and digoxin do not act directly on ERα but act as activators of the proteasome.
Altought the manuscript is globally well written and the results are of interest, there are many modifications that need to be adressed before publication.
General points
-The first criticism is the size of the figures that is really too small, particularly Figure 2 .
- There are too many supp results. Supp figures are supposed to show details of less importance. As an example (lane 560 : the paragraph describes only results from supp data), The main figures are not full so some supp figures can be moved to the main figures. In addition, tt is not appropriate to have a supplemental section for results and discussion, it makes the manuscript too disjointed.
Major points
-The authors can’t claim that the four cell lines used are assimilated to primary and metastatic breast cancer cells as they are genetically or pharmacolgically modified MCF7 cells. (Title and last sentence of the abstract).
-No information are given about the xenograft experiment, strain, numbers of fish used. Etc…
- Figure 1A and B are not very informative. I would like to see the results on cell proliferation and ERα degradation.
Figure 1B : How the authors explain that the IC50 is very high for ERα degradation compared to those of cell proliferation ?
- Why did the authors choose to use the same concentration of both GCs to perform transcriptomic analysis although their efficacy on cell proliferation, apotosis is not with the same ED50 ?
- Why did the authors focus in ZR75 but performed the transcriptomic analyses on MCF7 cells.
-Figure 3A :
-How the authors explain that in Y537S MCF7 Ouabain does not modify activities of the proteasome alghout ER ais degraded in this cell line.
-Figure 6D : the error bar are too big for the control, more fish must be used.
Minor points :
- In the abstract, CGs is not explained.
- Figure 1A and B: We do not see very well the differences between the different conditions. Please can the authors enlarge the scale?
- A modification should also be done in the supplemntal results, Sup Fig3B is described before Sup Figure 2N
- Figure Supp 1B should be independent of Supp Figure 1A, there is no link.
- Figure 5G: the schema should appear in the last figure.
- Figure 6A and 6B: Why do we see twice the same experiments with different results?
Author Response
Reviewer #1
By using a library of FDA approved drugs, the authors previously identified several hits able to down regulate the level of ER . In this manuscript the authors focused on 2 cardiac glycosides : ouabain and digoxin. They showed that these drugs prevent cell proliferation and trigger apoptosis of several BC cell lines. Moreover, they confirmed their results in fish xenografts. Mechanistically, the authors showed that ouabain and digoxin do not act directly on ERα but act as activators of the proteasome.
Altought the manuscript is globally well written and the results are of interest, there are many modifications that need to be adressed before publication.
General points
-The first criticism is the size of the figures that is really too small, particularly Figure 2.
Author response: We have now prepared a revised figure 3 in which we have eliminated panel A and B. This modification gave us room for enlarging panel C and D, which are now A and B. In addition, we made bigger each figure.
- There are too many supp results. Supp figures are supposed to show details of less importance. As an example (lane 560 : the paragraph describes only results from supp data), The main figures are not full so some supp figures can be moved to the main figures. In addition, tt is not appropriate to have a supplemental section for results and discussion, it makes the manuscript too disjointed.
Author response: We have now introduced most of the supplementary figures in the main text and included the text files in the main text in order to make the manuscript less disjoined.
Major points
-The authors can’t claim that the four cell lines used are assimilated to primary and metastatic breast cancer cells as they are genetically or pharmacolgically modified MCF7 cells. (Title and last sentence of the abstract).
Author response: Indeed, we do not want to claim that the cells used herein are assimilated to primary and metastatic breast cancer. We have always written in the text that these cells are models mimicking different types of primary and metastatic breast cancer. However, we corrected the title stressing the fact that we are using cells modelling primary and metastatic breast cancer and the relative parts in the text.
-No information are given about the xenograft experiment, strain, numbers of fish used. Etc…
Author response: We added the requested information in the material and method section. The experiments in Zebrafish xenografts was performed by the ZeClinics company as service. Tumor area was measured as ratio of the xenograft tumor in each used fish. Four fishes were used for the control groups, 9 fishes were used for the OU-treated group and 8 fishes were used for the Digo-treated groups. We added this information in the figure legend. Moreover, we added both in the method and in the relative figure legend that, upon reasonable request, we are available to share their final report, which includes all the requested details. In addition, the experiment was performed according to the methods published in reference #31, which we acknowledged in the reference section.
- Figure 1A and B are not very informative. I would like to see the results on cell proliferation and ERα degradation.
Author response: We have now prepared Supplementary figure 1 and 2 that include the growth curves, and the dose response curves for the ERα in all the cell line used and treated with OU and Digo.
Figure 1B: How the authors explain that the IC50 is very high for ERα degradation compared to those of cell proliferation?
Author response: We thank the reviewer for raising this point. However, it is important to point out that this is the case ONLY for Tam Res MCF-7 cells as in all other cell lines the two values are comparable. Notably, in this work, we report that digoxin activates the proteasome. In turn, most of ubiquitinated proteins are degraded as we observed with the specific assays. Consequently, the difference observed in Tam Res MCF-7 cell could indicate that while on one hand these cells are more resistant to the degradation of the receptor on the other hand, they could be more susceptible to the degradation of unknown critical proteins that regulate cell proliferation. This is therefore a very interesting point, but we feel this is out of the scope of the present work and would require specific investigation. In any case, finally, please note that the difference is however within the nanomolar range. We added the discussion of this point in page 19 line 962-966.
- Why did the authors choose to use the same concentration of both GCs to perform transcriptomic analysis although their efficacy on cell proliferation, apotosis is not with the same ED50?
Author response: We thank the reviewer for her/his comment. Initially, we decided to use same concentrations of drugs to eliminate any unspecific effects on MCF7 transcriptome due to diverse concentrations of the two drugs. Of course, we agree with the reviewer that the biological response to OU and DIGO can be slightly different due to different ED50 values. However, as shown in Figure 3, at least the transcriptional profile appeared to be almost identical which, for the purposes of our transcriptional analysis, make these results directly comparable. We added this point in the results section (page 8 line 396-397).
- Why did the authors focus in ZR75 but performed the transcriptomic analyses on MCF7 cells.
Author response: We choose to perform transcriptomic analyses in MCF-7 cells to be able to directly compare the effect of the two drugs in cell lines with a similar genetic background (i.e., Tam Res and Y537S). We added this point in the result section (page 8 line 396-397).
-Figure 3A:
-How the authors explain that in Y537S MCF7 Ouabain does not modify activities of the proteasome alghout ER ais degraded in this cell line.
Author response: Ouabain indeed induces the activation of the proteasome also in Y537S MCF-7 cells. In order to clarify this point, in addition to the heatmap depicted in revised figure 4, we have now added the relative panels in revised supplementary figure 5.
-Figure 6D: the error bar are too big for the control, more fish must be used.
Author response: Unfortunately, we cannot perform additional experiments in Zebrafish xenografts as we used ZeClinics as a company performing this service. In turn, they used the number of fishes that are adequate to perform the requested experiments (also for animal care purposes). As outlined above, we added the number of fishes used in the figure legend and briefly described the procedure used in the methods section. Their analysis reported significant differences between the CTR and treated groups, thus on this basis we concluded what they concluded: OU and Digo are effective also in a Zebrafish xenograft. Nonetheless, we added both in the method and in the relative figure caption that, upon reasonable request, we are available to share their final report.
Minor points:
In the abstract, CGs is not explained.
Author response: We have now corrected this point.
Figure 1A and B: We do not see very well the differences between the different conditions. Please can the authors enlarge the scale?
Author response: We have now updated the scales of panel A and B in the revised figure 1.
A modification should also be done in the supplemntal results, Sup Fig3B is described before Sup Figure 2N
Author response: We have now moved most of the supplementary figures in the main text as per previous request.
Figure Supp 1B should be independent of Supp Figure 1A, there is no link.
Author response: We have now separated the indicated panels in two independent figures.
Figure 5G: the schema should appear in the last figure.
Author response: The proposed model of the action of CGs in breast cancer cells is now indicated as figure 10, the last figure of the manuscript.
Figure 6A and 6B: Why do we see twice the same experiments with different results?
Author response: Figure 6A and 6B are now revised figure 9A and 9B. They are not the same experiment as they depict the synergistic effect of OU and Digo with Tam in both MCF-7 (9A) and ZR-75-1 cells (9B). Thus, they represent the same experiment but in two different cell lines.
Reviewer 2 Report
Claudia Busonero and co-authors presented the study devoted to the development of new approach to ER-containing breast cancer treatment. The authors showed that cardiac glycosides ouabain and digoxin activated proteasomes and ERα degradation and prevented proliferation of primary and metastatic breast cancer cells. On the base of the results obtained, the authors believe that these cardiac glycosides, being not such toxic as current anti-estrogen drugs, could be considered as appealing candidates in the treatment of primary and metastatic breast cancers. I think that MS is interesting and actual for specialists in this field and may be published in Cancers after minor revision.
1) In the title, the words “degrade the ERα” should be changed by “degradation of the ERα”, since ouabain and digoxin do not degrade the receptors but they activate the degradation of the receptors by proteasomes.
2) It is clear that in Fig. 5, panel E, OU should be changed by Digo.
3) The authors write about synergistic effects of OU and Tam, Digo and Tam. Unfortunately, they do not show effects OU or Digo alone (without Tam) in Fig. 6, panels A and B. Perhaps, the authors indicated Tam concentrations incorrectly.
4) In Fig. 2, panels A, B, C, D, the letters are very small, it is impossible to read them.
5) In Discussion, the authors should indicate that it is necessary to select (choose) concentrations of OU and Digo and control their effects on proteasomes at the development of drugs. Proteasome activation should be limited to avoid high activity level which is connected with cancer development. On the whole, investigators develop drugs with inhibiting effects on proteasome forms (https://doi.org/10.3390/cancers10100351). At the same time, for the treatment of ER dependent breast cancer, the limited proteasome activation is prospective.
Author Response
Reviewer #2
Claudia Busonero and co-authors presented the study devoted to the development of new approach to ER-containing breast cancer treatment. The authors showed that cardiac glycosides ouabain and digoxin activated proteasomes and ERα degradation and prevented proliferation of primary and metastatic breast cancer cells. On the base of the results obtained, the authors believe that these cardiac glycosides, being not such toxic as current anti-estrogen drugs, could be considered as appealing candidates in the treatment of primary and metastatic breast cancers. I think that MS is interesting and actual for specialists in this field and may be published in Cancers after minor revision.
Author response: we thank the reviewer for this comment.
1) In the title, the words “degrade the ERα” should be changed by “degradation of the ERα”, since ouabain and digoxin do not degrade the receptors but they activate the degradation of the receptors by proteasomes.
Author response: We agree with this reviewer and changed the title accordingly.
2) It is clear that in Fig. 5, panel E, OU should be changed by Digo.
Author response: We thank the reviewer for pointing out this mistake, which we corrected in the revised figure 5 panel E.
3) The authors write about synergistic effects of OU and Tam, Digo and Tam. Unfortunately, they do not show effects OU or Digo alone (without Tam) in Fig. 6, panels A and B. Perhaps, the authors indicated Tam concentrations incorrectly.
Author response: We thank the reviewer for pointing out this mistake, which we corrected in the revised figure 6A and 6B. Indeed, we indicated Tam concentrations incorrectly.
4) In Fig. 2, panels A, B, C, D, the letters are very small, it is impossible to read them.
Author response: We have now prepared a revised figure 2 in which we have eliminated panel A and B. This modification gave us room for enlarging panel C and D that are now panel A and B.
5) In Discussion, the authors should indicate that it is necessary to select (choose) concentrations of OU and Digo and control their effects on proteasomes at the development of drugs. Proteasome activation should be limited to avoid high activity level which is connected with cancer development. On the whole, investigators develop drugs with inhibiting effects on proteasome forms (https://doi.org/10.3390/cancers10100351). At the same time, for the treatment of ER dependent breast cancer, the limited proteasome activation is prospective
Author response: We thank the review for raising this point. We have now added a paragraph in the discussion section where we suggest caution in the use of CGs for the treatment of any pathology as they activate the proteasome. Additionally, we stressed however the fact that in BC their use can be a valuable strategy to treat the disease. We also cited the paper indicated by this Referee.
Round 2
Reviewer 1 Report
The authors addressed all my concerns.